# Remote solid cancers rewire hepatic nitrogen metabolism via host nicotinamide-*N*-methyltransferase

Rin Mizuno[1,2], Hiroaki Hojo[1,3,4], Masatomo Takahashi [5], Soshiro Kashio[6], Sora Enya[3,4], Motonao Nakao[5], Riyo Konishi[1], Mayuko Yoda[1], Ayano Harata[1], Junzo Hamanishi [2], Hiroshi Kawamoto [7], Masaki Mandai[2], Yutaka Suzuki [8], Masayuki Miura [6], Takeshi Bamba [5], Yoshihiro Izumi [5] & Shinpei Kawaoka [1,3,4,9 ✉]

Cancers disrupt host homeostasis in various manners but the identity of host factors underlying such disruption remains largely unknown. Here we show that nicotinamide-*N*-methyltransferase (NNMT) is a host factor that mediates metabolic dysfunction in the livers of cancer-bearing mice. Multiple solid cancers distantly increase expression of *Nnmt* and its product 1-methylnicotinamide (MNAM) in the liver. Multi-omics analyses reveal suppression of the urea cycle accompanied by accumulation of amino acids, and enhancement of uracil biogenesis in the livers of cancer-bearing mice. Importantly, genetic deletion of *Nnmt* leads to alleviation of these metabolic abnormalities, and buffers cancer-dependent weight loss and reduction of the voluntary wheel-running activity. Our data also demonstrate that MNAM is capable of affecting urea cycle metabolites in the liver. These results suggest that cancers up-regulate the hepatic NNMT pathway to rewire liver metabolism towards uracil biogenesis rather than nitrogen disposal via the urea cycle, thereby disrupting host homeostasis.

[1] Inter-Organ Communication Research Team, Institute for Life and Medical Sciences, Kyoto University, Kyoto 606-8507, Japan. [2] Department of Gynecology and Obstetrics, Kyoto University Graduate School of Medicine, Kyoto 606-8507, Japan. [3] The Thomas N. Sato BioMEC-X Laboratories, Advanced Telecommunications Research Institute International (ATR), Kyoto 619-0237, Japan. [4] ERATO Sato Live Bio-forecasting Project, Japan Science and Technology Agency (JST), Kyoto 619-0237, Japan. [5] Division of Metabolomics, Research Center for Transomics Medicine, Medical Institute of Bioregulation, Kyushu University, Fukuoka 812-8582, Japan. [6] Department of Genetics, Graduate School of Pharmaceutical Sciences, The University of Tokyo, Tokyo 113-0033, Japan. [7] Laboratory of Immunology, Institute for Frontier Life and Medical Sciences, Kyoto University, Kyoto 606-8507, Japan. [8] Graduate School of Frontier Science, The University of Tokyo, Chiba 277-8562, Japan. [9] Department of Integrative Bioanalytics, Institute of Development, Aging and Cancer (IDAC), Tohoku University, Sendai 980-8575, Japan. ✉email: kawaokashinpei@gmail.com

C ancers adversely affect various host organs at a body-wide level[1,2]. In the presence of solid cancers, for example, adipose tissues exhibit enhanced lipolysis and thus degenerate (i.e., adipose atrophy)[3]. Metabolism in the liver is also adversely affected in cancer-bearing organisms despite having been less studied than the adipose tissues in this context[4–8]. Such metabolic abnormalities, often accompanied by chronic inflammation, systemically disrupt host homeostasis, and lead to loss of body weight, reduced quality of life, impaired tolerance to standard anti-cancer therapies, and ultimately worsened survival[1,2]. This multifactorial and systemic phenomenon is clinically recognized as cancer cachexia, a life-threatening syndrome that more than 50% of advanced cancer patients unfortunately suffer[9].

Nicotinamide-*N*-methyltransferase (NNMT) is an enzyme that transfers a methyl group from *S*-adenosyl methionine (SAM) to nicotinamide (NAM)[10]. This reaction produces *S*-adenosyl homocysteine (SAH) and 1-methylnicotinamide (MNAM). In mice, NNMT is abundantly expressed in the liver and adipose tissues[11,12]. NNMT was originally recognized as merely an NAM clearance enzyme that discards NAM as MNAM since MNAM is excreted in the urine[10]. However, this view has been challenged by recent important studies that demonstrate the significant roles of NNMT in obesity[11], metabolism[12,13], cancers[14,15], immunity[16], and lifespan[17].

Despite initially being characterized as metabolically inert in the past, MNAM is now thought to harbor various biological functions[11,12,16–20]. For example, it is known that MNAM binds and stabilizes SIRT1 thereby contributing to hepatic nutrient metabolism[12]. MNAM is also considered as an anti-inflammatory agent[20]. In addition, MNAM is metabolized by aldehyde oxidase (AOX), leading to the production of reactive oxygen species (ROS). In *Caenorhabditis elegans* (*C. elegans*), ROS generated by this reaction is believed to prolong lifespan[17].

Whether NNMT and MNAM are involved in cancer-induced host pathophysiology is currently unknown. Yet, there is evidence implicating NNMT as such a factor. The first indication was reported in 1998 showing that transplantation of colon cancers into mice increased the enzymatic activity of NNMT in the liver[21]. Supporting this, our transcriptome experiments demonstrated strongly induced expression of *Nnmt* in the liver after transplantation of breast cancers[6]. A similar trend was also seen in the livers of mice bearing genetically induced lung cancers[4]. Thus, a series of non-liver solid cancers induced NNMT expression in the liver. How the NNMT induction in the liver affects liver physiology in the cancer-bearing condition remained to be revealed.

In the current study, utilizing mouse genetics and multi-omics analyses, we establish NNMT as a host factor that promotes metabolic dysfunction of the liver in cancer-bearing animals. The multi-omics map we construct here provides a comprehensive picture of cancer-induced abnormalities in liver metabolism. Moreover, our data reveal which abnormalities depend on host NNMT (and which do not). This study not only unravels the roles of NNMT in cancer-induced host pathophysiology but also offers a basis to study the complex systemic syndrome that severely impacts patients with incurable cancers.

## Results

**Solid cancers upregulate *Nnmt* expression in the liver**. Our previous research demonstrated that expression of *Nnmt* in the liver is increased upon transplantation of 4T1 breast cancer[6]. We confirmed this observation using quantitative polymerase chain reaction (qPCR) (Fig. 1a, b and Supplementary Data 1). The degree of induction was correlated to the duration of time following transplantation (i.e., Day 7 < Day 14). Induction of hepatic

*Nnmt* was not restricted to the 4T1 breast cancer model as colon cancer (Colon26), ovarian cancer (ID8-F3), and lung cancer (LLC) all increased hepatic *Nnmt* mRNAs in varying degrees (Fig. 1c). The Colon26 data were consistent with the previous finding[21]. In addition, the *Nnmt* induction was detected in the livers of a genetically induced lung cancer model (Supplementary Fig. 1a)[4], demonstrating that the induction was not a result of transplantation. Therefore, upregulation of hepatic NNMT by solid cancers appeared a general phenomenon. In this study, we largely focused on the 4T1 breast cancer model since 4T1 was the most prominent inducer of hepatic *Nnmt*.

Next, we investigated the effects of this *Nnmt* induction on NNMT-related metabolites. NNMT transfers a methyl group from *S*-adenosyl methionine (SAM) to nicotinamide (NAM), producing *S*-adenosyl homocysteine (SAH) and 1-methylnicotinamide (MNAM)[10]. Liquid chromatography coupled with tandem mass spectrometry (LC-MS/MS) demonstrated that the steady-state amount of MNAM was increased in the livers of 4T1-bearing mice (Fig. 1d). On the other hand, 4T1 breast cancer did not affect NAM, SAM, and SAH in the liver (Fig. 1d). In addition, it is known that MNAM is further metabolized into me4PY (*N*-methyl-4-pyridone-3-carboxamide) and me2PY (*N*-methyl-2-pyridone-5-carboxamide) by aldehyde oxidase (AOX)[10,22,23]. Yet, 4T1 transplantation did not affect these two metabolites (Supplementary Fig. 1b).

We next sought to address how 4T1 breast cancer increases *Nnmt* expression in hepatocytes. For this purpose, we exploited an approach that uses supernatant of 4T1 culture (4T1-conditioned media): we treated AML12 primary hepatocyte cells with the 4T1-conditioned media and examined the expression of *Nnmt*. We found that 4T1-conditioned media was capable of up-regulating *Nnmt* expression in AML12 (Fig. 1e). This was accompanied by a concomitant increase in MNAM and a decrease in SAM (Fig. 1f). 4T1 expresses a variety of cytokines and hormones in vitro including TNFα as reported by others and also confirmed by us (Supplementary Fig. 1c)[24]. Our data demonstrated that TNFα was able to increase *Nnmt* mRNAs and MNAM (Fig. 1g, h). These data collectively suggested that 4T1 breast cancer activates the hepatic NNMT pathway at least in part via soluble factors such as TNFα.

**_Nnmt_ deletion abolishes MNAM and accumulates SAM in a cancer-bearing condition**. Changes in *Nnmt* expression in the liver prompted us to investigate the role of *Nnmt* in cancer-induced abnormalities in this organ. To address this, we generated *Nnmt* knockout (KO) mice using the CRISPR-Cas9 technique (Fig. 2a, b)[25]. We obtained mice that harbor a 25-base pair (bp) deletion. This resulted in a premature stop codon in the domain critical for the enzymatic activity of NNMT. In this manuscript, we simply refer to this allele as *Nnmt* KO.

To characterize *Nnmt* KO mice, we measured the amounts of NAM, SAM, MNAM, and SAH in the liver. In the *Nnmt* KO livers, only MNAM was strongly affected by *Nnmt* KO; MNAM was almost nearly abolished in *Nnmt* KO mice (Fig. 2c). This indicated that MNAM is generated solely by NNMT. Moreover, me4PY and me2PY were depleted by *Nnmt* KO (Supplementary Fig. 2a). These were in contrast to NAM, SAM, and SAH were all unaffected by loss of NNMT function. In addition, nicotinamide adenine dinucleotide (NAD) was also unaffected in the liver (Supplementary Fig. 2b). It thus appeared that, in a cancer-free condition, the contribution of NNMT in regulating NAM, SAM, SAH, and NAD is not prominent.

We then measured the amounts of NNMT-related metabolites in 4T1 breast cancer-bearing mice. In contrast to sham-treated mice, *Nnmt* KO accumulated SAM within the livers of 4T1-

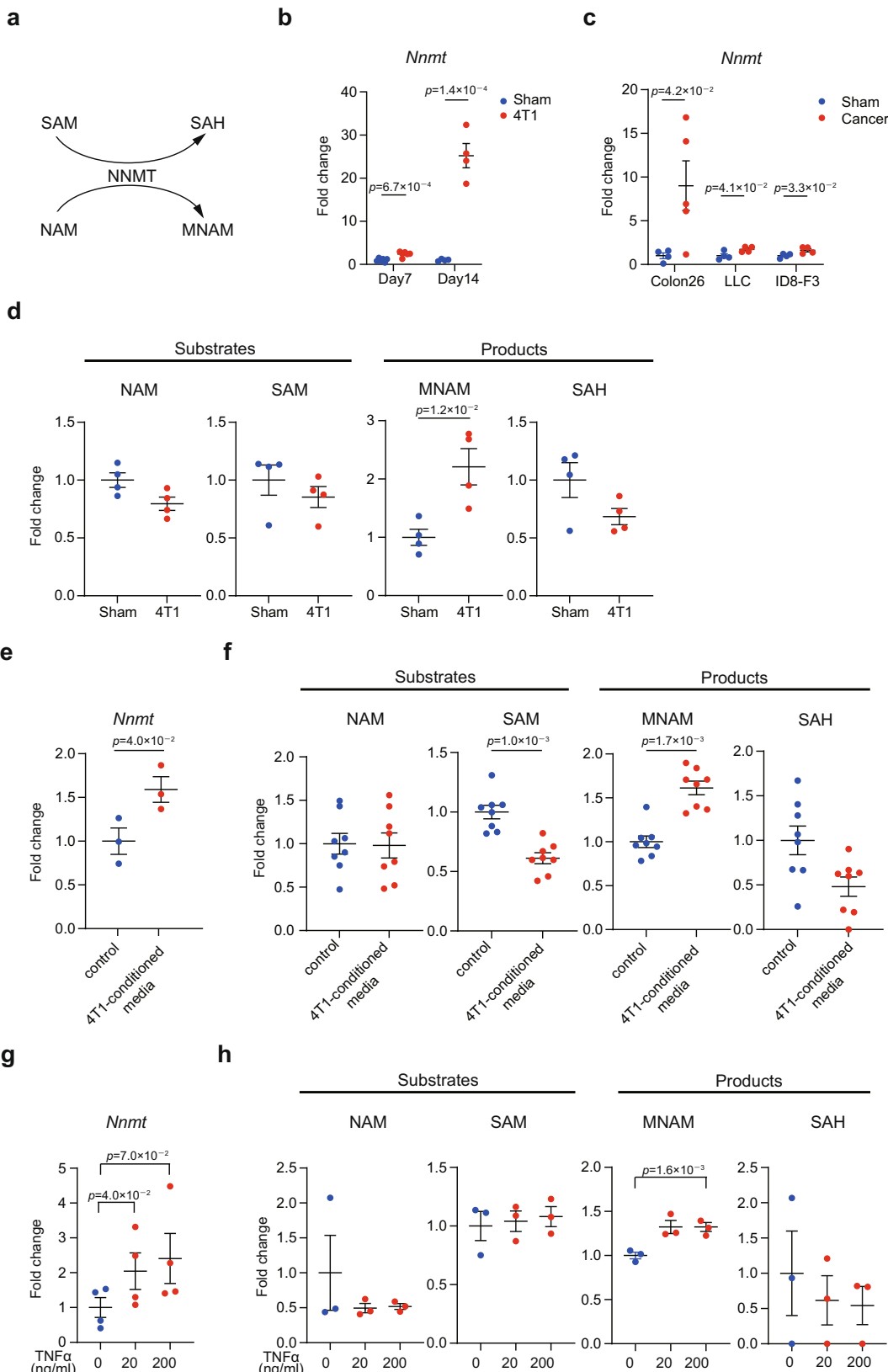

bearing mice (Fig. 2d) accompanied by a consequent increase in the SAM/SAH ratio (Fig. 2e), indicative of disrupted methyl-donor balance. It is known that glycine *N*-methyltransferase (GNMT) is the major consumer of SAM in the liver and that *Gnmt* KO accumulates SAM[26,27]. Gene expression analysis showed that expression of *Gnmt* was decreased in the 4T1-

bearing condition (Supplementary Fig. 2c). These data suggested that NNMT affects the amount of SAM in a hypo-*Gnmt*-like condition. This was in line with a recent paper showing that *Nnmt* RNAi results in an increase of SAM specifically upon *Gnmt* RNAi[27]. We also noted that the loss of *Nnmt* affected NAD in the 4T1 breast cancer-bearing condition (Supplementary Fig. 2d).

**Fig. 1 Solid cancers upregulate *Nnmt* expression in the liver. a** The biochemical property of NNMT. **b** qPCR analysis for *Nnmt* in the livers of 4T1-bearing mice on 7 and 14 days after transplantation. $n = 6$ for D7, $n = 4$ for D14. **c** qPCR analysis for *Nnmt* in the livers of the colon (Colon26), lung (LLC), and ovarian (ID8-F3) cancer-bearing mice. The livers were collected 14 days after transplantation in the colon26 and LLC models, and 42 days after transplantation in the ID8-F3 model. $n = 4$ for sham-operated mice, LLC-bearing mice, and ID8-bearing mice. $n = 5$ for Colon26-bearing mice. **d** Liquid chromatography with tandem mass spectrometry (LC-MS/MS) analysis for nicotinamide (NAM), S-adenosyl-methionine (SAM), 1-methylnicotinamide (MNAM), and S-adenosyl-homocysteine (SAH) in the livers of sham-operated and 4T1-bearing mice. $n = 4$. **e** qPCR analysis for *Nnmt* in a primary hepatocyte culture cell line AML12 treated with the 4T1-conditioned medium determined by qPCR. $n = 3$. **f** LC-MS/MS analysis for the NNMT-related metabolites in AML12 treated with the 4T1-conditioned medium. $n = 8$. **g** qPCR analysis to reveal the effect of TNFα on *Nnmt* expression in AML12. $n = 4$. **h** LC-MS/MS analysis for the NNMT-related metabolites in AML12 treated with TNFα. $n = 3$. The exact $p$ values are shown (unpaired two-tailed Student's $t$-test in **b**–**d** and paired two-tailed Student's $t$-test in **e**–**h**). Averaged fold change data normalized to the control groups are presented as the mean ± SEM. Source data are provided as a source data file.

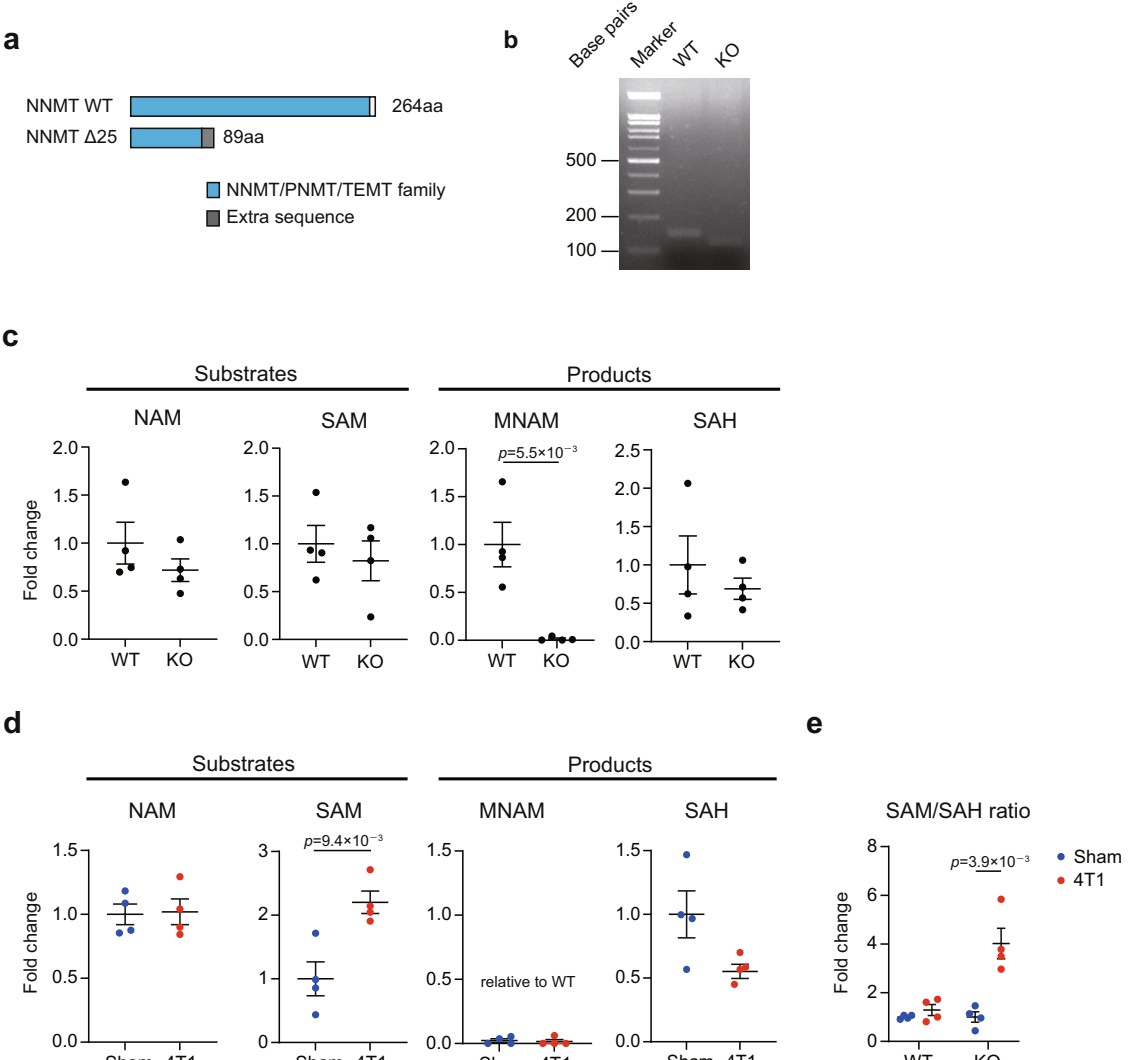

**Fig. 2 *Nnmt* KO abolishes MNAM and accumulates SAM in a cancer-bearing condition. a** The *Nnmt* KO (Δ25 allele) was generated by the CRISPR-Cas9 technique in this study. **b** The representative picture for genotyping BALB/c *Nnmt* KO mice. **c** LC-MS/MS analysis for the NNMT-related metabolites in the livers of WT and *Nnmt* KO mice. $n = 4$. **d** LC-MS/MS analysis for the NNMT-related metabolites in the livers of *Nnmt* KO mice in the 4T1 transplantation experiments. $n = 4$. **e** The SAM/SAH ratio of sham-operated and 4T1-bearing WT and *Nnmt* KO mice. $n = 4$. **c**–**e** The exact $p$ values are shown (unpaired two-tailed Student's $t$-test). Averaged fold change data are presented as the mean ± SEM. Source data are provided as a source data file.

4T1 transplantation severely depleted NAD in the liver. *Nnmt* KO partially buffered the cancer-induced NAD depletion.

Importantly, owing to the lack of MNAM in *Nnmt* KO, the increase of MNAM by 4T1 breast cancer in the livers of WT was completely canceled (Figs. 1d, 2d). Thus, in 4T1 breast cancer-bearing mice, *Nnmt* KO resulted in the accumulation of SAM while canceling the anticipated increase of MNAM in the liver.

***Nnmt* deletion buffers multi-omics changes caused by 4T1 breast cancer.** On the basis of the data presented in Figs. 1, 2, we expected that NNMT might play roles in cancer-induced abnormalities in the liver via regulation of the methyl-donor balance and/or MNAM. To test this idea, we performed multi-omics analyses (transcriptome and metabolome) on the livers of 4T1 breast cancer-bearing mice.

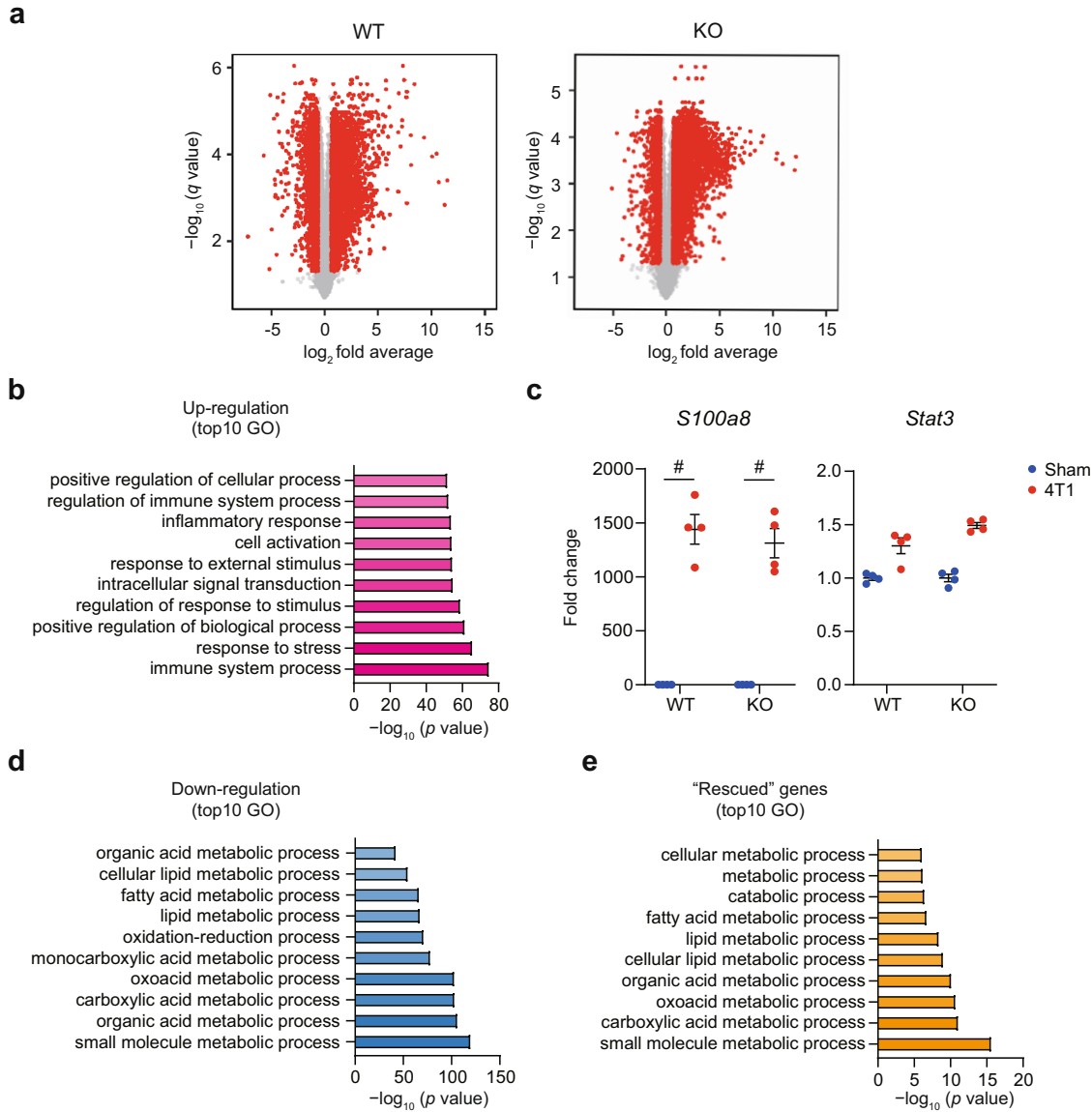

**Fig. 3 *Nnmt* deletion buffers transcriptome changes caused by 4T1. a** RNA-seq experiments for the livers of sham-operated mice and 4T1-bearing mice in WT and *Nnmt* KO (14 days after 4T1 transplantation). Volcano plots ($\log_2$ fold average (4T1/sham) versus $-\log_{10}$ ($q$ value)) of WT (left) and *Nnmt* KO (right) are shown. Genes showing more than 1.5-fold change with $q < 0.05$ are highlighted in red. $n = 4$. **b** Gene ontology analysis (g:Profier) for genes significantly elevated in the livers of 4T1-bearing mice. Adjusted enrichment $p$ values obtained from g:Profier are shown. **c** RNA-seq results of representative upregulated genes *S100a8* and *Stat3*. Averaged fold change data normalized to the sham group in each genotype are presented as the mean ± SEM. #; more than 1.5-fold change with $q < 0.05$. $n = 4$. **d** Gene ontology analysis (g:Profier) for genes decreased in the livers of 4T1-bearing mice. Adjusted enrichment $p$ values obtained from g:Profier are shown. **e** Gene Ontology analysis for genes whose downregulation was rescued by *Nnmt* KO in the 4T1-bearing condition. Adjusted enrichment $p$ values obtained from g: Profier are shown. Source data are provided as a source data file.

Transcriptome analyses revealed that 4T1 breast cancer affected global gene expression in the liver. On 14 days after transplantation, 2933 genes were upregulated while 1921 genes were downregulated amongst the 13,921 monitored genes (Fig. 3a; >1.5-fold change with $q < 0.05$, and Supplementary Data 2). As shown in our previous publication[6], elevated genes mostly represented inflammatory signatures (Fig. 3b). For example, *S100a8*, a gene expressed in monocytes, was strongly elevated, suggesting that monocytes had infiltrated the liver (Fig. 3c). In addition, it appeared that the cancer state activated the IL-6-STAT3 signaling axis. (Fig. 3c). Many decreased genes were involved in metabolism (Fig. 3d), suggesting that 4T1 breast cancer affected liver metabolism. During this analysis, we found that *Aox1* and *Aox3* were downregulated in the livers of 4T1-bearing mice

(Supplementary Fig. 3a). As described earlier, AOX proteins metabolize MNAM into me4PY and me2PY[10,22,23]. These together suggested that 4T1 breast cancers increase MNAM not only via NNMT upregulation but also via downregulation of *Aox* genes (i.e., suppression of MNAM degradation).

When the same experiment was done on *Nnmt* KO mice, 534 upregulated genes and 1002 genes showing downregulation in WT mice were normalized in the livers of *Nnmt* KO mice (Fig. 3a; did not meet either >1.5-fold change or $q < 0.05$). Such amelioration of gene regulation was not extended to inflammatory genes indicating that *Nnmt* was dispensable for cancer-induced liver inflammation (Fig. 3c). Gene ontology against genes downregulated in WT but not in *Nnmt* KO mice revealed that many of these genes participated in the metabolic processes in the

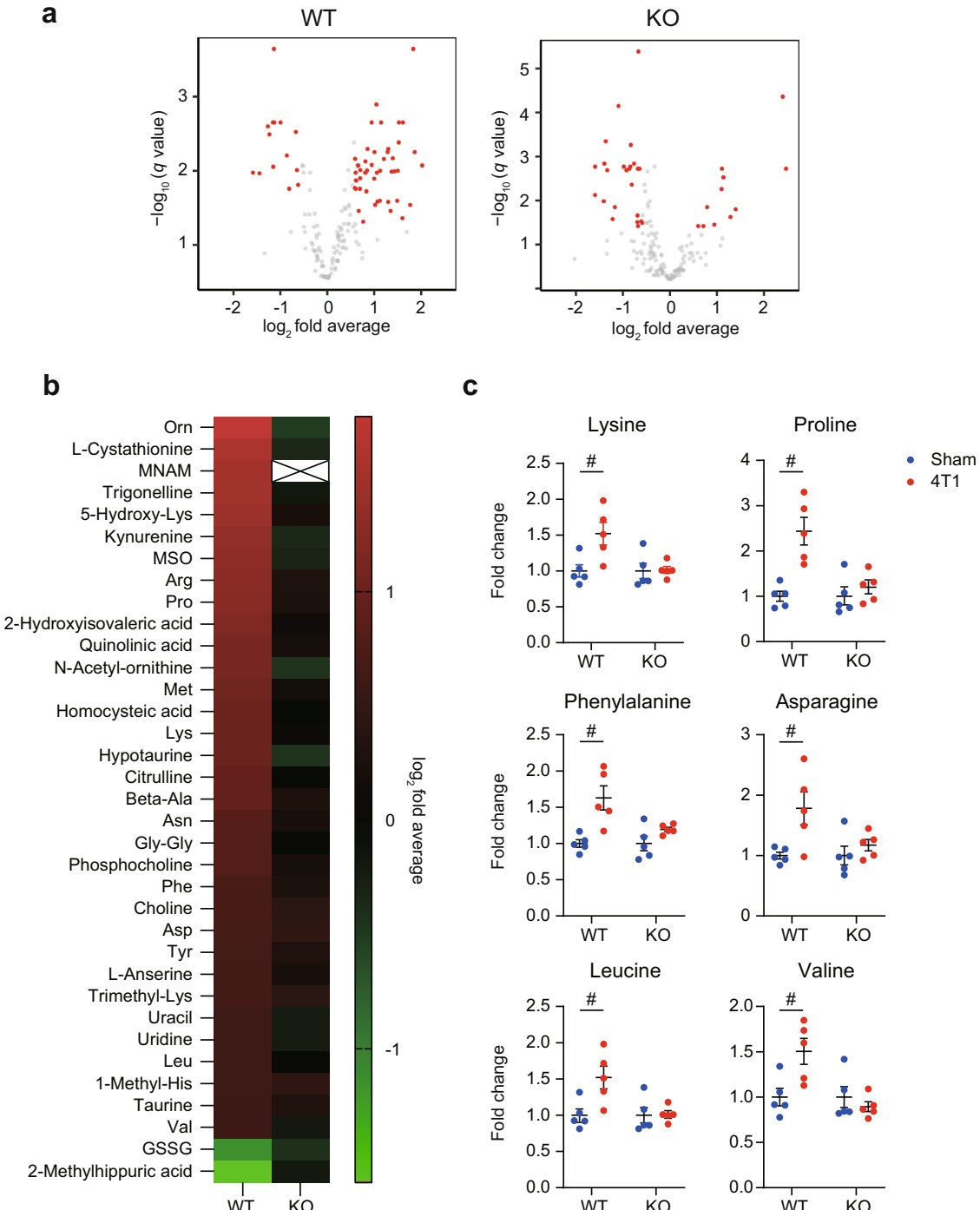

**Fig. 4 *Nnmt* deletion buffers metabolic changes caused by 4T1. a** Metabolome experiments for the livers of sham-operated mice and 4T1-bearing mice in WT and *Nnmt* KO (14 days after 4T1 transplantation). Volcano plots (log$_2$ fold average (4T1/sham) versus –log$_{10}$ (*q* value)) of WT (left) and *Nnmt* KO (right) are shown. Metabolites showing more than 1.5-fold change with *q* < 0.05 are highlighted in red. *n* = 5. **b** Heatmap representation of metabolites that are significantly affected in WT (left column) but not in *Nnmt* KO (right column). *n* = 5. **c** Representative plots of "rescued" metabolites are shown in **b**. Data from six representative amino acids are shown. Averaged fold change data normalized to the sham group in each genotype are presented as the mean ± SEM. #; more than 1.5-fold change with *q* < 0.05. *n* = 5. Source data are provided as a source data file.

liver (Fig. 3e). Thus, the transcriptome data might reflect the amelioration of 4T1 breast cancer-induced metabolic dysfunction by *Nnmt* disruption. In a healthy condition (in non-tumor bearing mice), the effects of *Nnmt* KO on liver gene expression appeared relatively small (Supplementary Fig. 3b).

The transcriptome data further prompted us to conduct more detailed metabolome analyses by measuring steady-state amounts

of 174 metabolites. As shown in Fig. 4a, we found that 4T1 breast cancer affected various metabolites in the liver. Among 174 metabolites, 50 were elevated while 14 were reduced (Fig. 4a; >1.5-fold change with *q* < 0.05, and Supplementary Data 3) in WT. For example, the accumulation of a set of amino acids such as arginine was notable (Figs. 4, 5). Other amino acids lysine, phenylalanine, asparagine, aspartate, and so on, were elevated. A

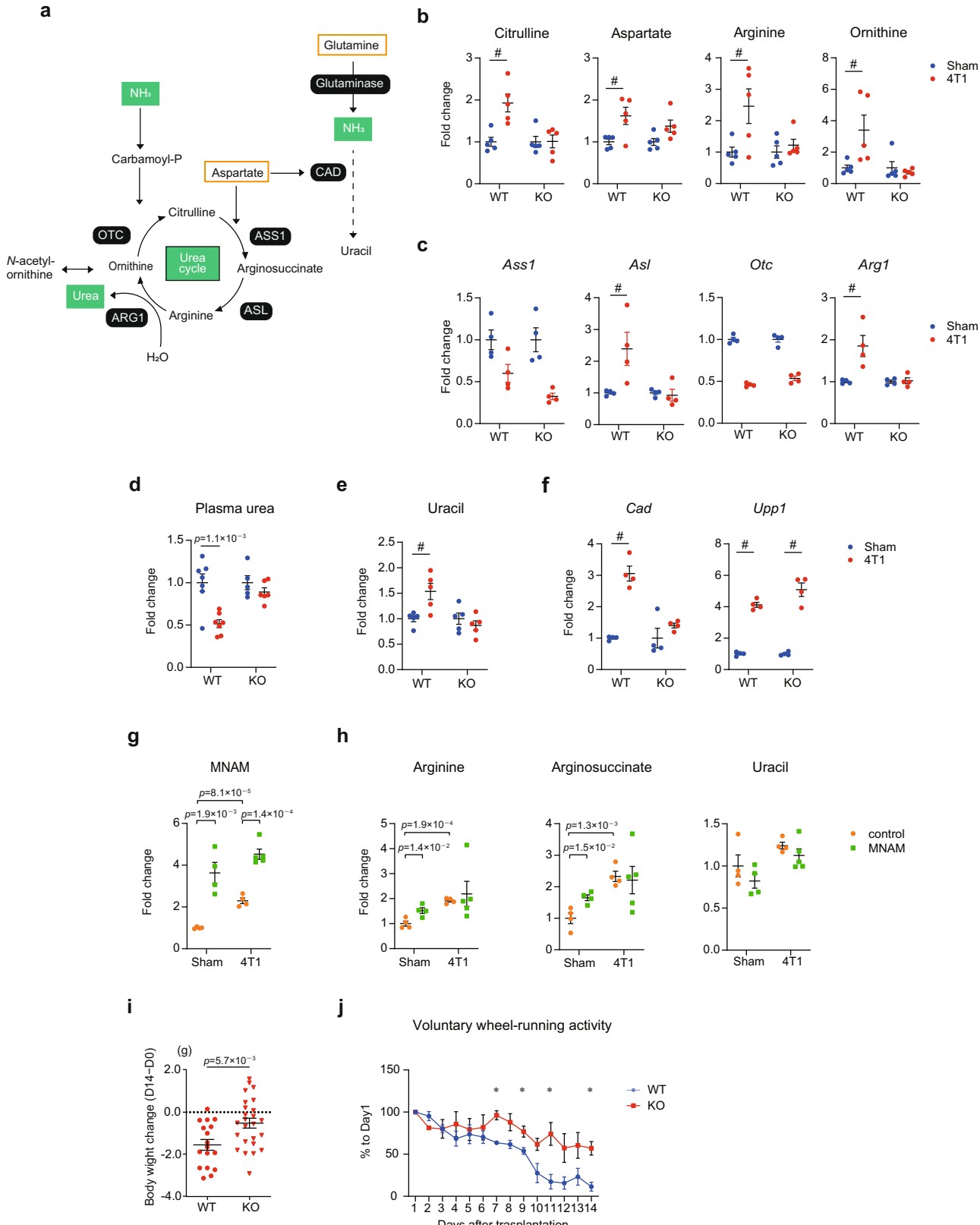

dipeptide, anserine, was also accumulated. These increases suggested several possibilities: increased protein degradation, an accelerated flux of amino acids into the liver, and/or defective amino acid degradation in the liver. Alterations of some of these metabolites such as lysine have been reported in other murine cancer models and even in human cachexia patients, lending

credence to our models and measurements[28,29]. In addition to amino acids, nucleic acids such as uracil were increased in our datasets. Metabolites and genes in the glycolysis pathway were mostly decreased (Supplementary Fig. 4a–c). These data together with transcriptome data indicated metabolic dysfunction in the livers of 4T1-bearing mice.

**Fig. 5 NNMT is required for dysregulation of the urea cycle and uracil biogenesis. a** The urea cycle. The part where the urea cycle is linked to uracil biogenesis is also shown. Nitrogen sources (aspartate and glutamine) are highlighted in orange. See also Supplementary Fig. 6a for more details. **b** LC-MS/MS analyses for the urea cycle metabolites and its derivative in the livers. $n = 5$. **c** RNA-seq measurements for genes encoding urea cycle enzymes in the livers. $n = 4$. **d** The plasma urea level was measured 14 days after 4T1 transplantation. The exact $p$ value is shown (unpaired two-tailed Student's $t$-test). Averaged fold change data normalized to the sham group in each genotype are presented as the mean ± SEM. $n = 7$ for WT, $n = 5$ for sham-operated *Nnmt* KO mice, $n = 6$ for 4T1-bearing *Nnmt* KO mice. **e** LC-MS/MS analysis for uracil in the livers. $n = 5$. **f** RNA-seq measurements for *Cad* and *Upp1*. $n = 4$. **g** LC-MS/MS analysis for MNAM in the livers from mice subjected to daily 250 mg/kg MNAM injection for 12 days in the absence (sham) and presence (4T1) of 4T1 cancers. $n = 5$ for 4T1-bearing MNAM treated mice. $n = 4$ for the other three experimental groups. **h** LC-MS/MS analyses for arginine, arginosuccinate, and uracil in the livers. $n = 5$ for 4T1-bearing MNAM treated mice. $n = 4$ for the other three experimental groups. **i** Body weight changes measured on D14 after 4T1 transplantation in WT and *Nnmt* KO. The weights are calculated as total body weight − cancer mass. $n = 17$ for WT, $n = 25$ for *Nnmt* KO. Data were represented as the mean ± SEM. The exact $p$ value is shown (unpaired two-tailed Student's $t$-test). **j** The voluntary wheel-running activity was measured 1–14 days after 4T1 transplantation in WT and *Nnmt* KO. The number of rotations on each day is shown as % to that on Day1 in each genotype. $n = 3$. Data were presented as the mean ± SEM. *$p < 0.05$, unpaired two-tailed Student's $t$-test. The exact $p$ values are $4.5 \times 10^{-3}$ (Day 7), $3.9 \times 10^{-2}$ (Day 9), $2.4 \times 10^{-2}$ (Day 11), and $9.0 \times 10^{-3}$ (Day 14). **b, c, e, f** Averaged fold change data normalized to the sham group in each genotype are presented as the mean ± SEM. #; >1.5-fold change with $q < 0.05$. **g, h** Averaged fold change data normalized to the sham-operated control mice are presented as the mean ± SEM. The exact $p$ values are shown (unpaired two-tailed Student's $t$-test). Source data are provided as a source data file.

We found that *Nnmt* KO reduced the number of cancer-affected metabolites, in particular those that accumulated (Fig. 4a, b). Among the 50 increased metabolites, 33 were not significantly elevated in the livers of *Nnmt* KO mice. As exemplified in Fig. 4c, the accumulation of a series of amino acids lysine, proline, phenylalanine, and so on, was rescued. These data suggested that NNMT deletion ameliorated defective amino acid metabolism. In contrast, the downregulation of glycolysis metabolites was unaffected by *Nnmt* KO (Supplementary Fig. 4b, c).

*Nnmt* KO affected the relatively small number of metabolites (six metabolites) in mice not bearing cancer (Supplementary Fig. 5a). Other than MNAM, the most prominently affected metabolite in the sham group was trigonelline (Supplementary Fig. 5a, b). Trigonelline is a methylated form of nicotinic acids[30]. It seemed plausible that *Nnmt* disruption directs SAM to synthesize trigonelline. In other words, trigonelline might function as a previously unrecognized recipient for a methyl group normally reserved for MNAM production but now available in the absence of *Nnmt* (Supplementary Fig. 5c). The increase of trigonelline might explain why *Nnmt* KO in the healthy state did not elevate SAM. Trigonelline also accumulated in the livers of cancer-bearing WT mice that also showed elevated *Nnmt* and decreased *Gnmt*. SAM appeared unaltered in this condition, suggesting that upregulation of MNAM and trigonelline biogenesis might maintain SAM homeostasis when *Gnmt* was reduced. This suggested that trigonelline biogenesis is a factor in the maintenance of the methyl-donor balance in the liver (Supplementary Fig. 5c). These results together suggest that NNMT plays roles in liver metabolism, particularly in the cancer-bearing condition.

**NNMT is required for dysregulation of the urea cycle and uracil biogenesis**. To deeper understand the biological roles of NNMT in 4T1 breast cancer-induced abnormalities in liver metabolism, we tried to identify metabolic pathways that were regulated by NNMT in the 4T1-bearing condition with the aid of integrating transcriptome and metabolome datasets. To this end, we mapped the altered metabolites and genes utilizing the Kyoto encyclopedia of genes and genomes (KEGGs)[31]. Based on these datasets, NNMT appeared to be involved in dysregulation of the urea cycle and in enhanced uracil biogenesis in the 4T1 breast cancer-bearing condition (Fig. 5).

The urea cycle is the pathway that converts toxic ammonia into non-toxic urea (i.e., dissipates nitrogen from a tissue) (Fig. 5a and Supplementary Fig. 6a)[32,33]. Thus, the urea cycle is critical for nitrogen homeostasis. We noted that a set of amino acids

increased by 4T1 breast cancer in the liver was mapped to this important metabolic pathway. In WT, 4T1 breast cancer elevated the amount of citrulline, aspartate, arginine, and ornithine in the liver (Fig. 5b). These changes were accompanied by alterations in the steady-state levels of mRNAs encoding the urea cycle enzymes (Fig. 5c). Expression of *Arginase1* (*Arg1*) and *Arginosuccinate lyase* (*Asl*) was elevated by 4T1 breast cancer (Fig. 5c) while the mRNA levels of *Ornithine carbamoyltransferase* (*Otc*) and *Arginosuccinate synthase 1* (*Ass1*) were reduced. The upregulation of *Asl* and *Arg1* correlated with the accumulation of their products (arginine and ornithine, respectively). In addition, 4T1 transplantation reduced the plasma urea level (Fig. 5d). On the basis of these findings, we concluded that 4T1 breast cancer complexly rewired the urea cycle to reduce its overall activity in the host liver.

There is an emerging connection reported between the disruption of the urea cycle and uracil biogenesis (Supplementary Fig. 6a)[32–34]. This link can be seen in an *Ass1* deficiency that shunts aspartate to uracil synthesis rather than the urea cycle (Fig. 5a)[32–34]. In fact, a previous study showed that an *Ass1* deficiency increases uracil in cancer cells[34]. The increase of uracil in *Ass1*-deficient cells requires *carbamoyl-phosphate synthase 2, aspartate transcarbamylase, and dihydroorotase* (*Cad*), which encodes an enzyme that directs aspartate for uracil production[34]. *Cad* is a marker of uracil biogenesis. Of note, aspartate and uracil were simultaneously elevated in the livers of 4T1 breast cancer-bearing mice, accompanied by upregulation of *Cad* and by downregulation of *Ass1* (Figs. 5b, c, e, f, respectively). The accumulation of uracil could also be attributed to enhanced catabolism of uridine. Uridine phosphorylase (UPP1) is essential in metabolizing uridine into uracil and the expression of *Upp1* is a marker for uridine degradation and uracil production[35]. We found that 4T1 breast cancer increased *Upp1* expression in the liver (Fig. 5f), implying that there is an increased flux of uridine to uracil in the 4T1 breast cancer-bearing condition. Collectively, our data demonstrated that urea cycle dysregulation caused by cancers was associated with enhanced uracil biogenesis. These results suggested that cancers suppress nitrogen disposal and instead increase uracil biogenesis in the host liver.

We then looked into the effects of *Nnmt* KO in the urea cycle dysregulation and uracil biogenesis. We found that *Nnmt* KO ameliorated the alteration in the urea cycle and uracil biogenesis (Fig. 5b, e): the amounts of arginine, citrulline, ornithine, and uracil did not significantly differ between the sham group and the 4T1 breast cancer-bearing group in the absence of *Nnmt*. These were consistent with gene expression profiles whereby *Nnmt* KO suppressed the upregulation of *Asl* and *Arg1* in the 4T1 breast

cancer-bearing condition. Importantly, the decrease of plasma urea was almost completely normalized in *Nnmt* KO mice (Fig. 5d), suggestive of amelioration in the urea cycle dysregulation at the systemic level. We validated these results with the Lewis Lung Carcinoma (LLC) model and B6 *Nnmt* KO (Supplementary Fig. 6b–e). We found a trend that LLC reduced the plasma urea level ($p = 0.08$) 14 days after transplantation, despite being less pronounced than the changes seen in 4T1 breast cancer-bearing mice (Fig. 5d and Supplementary Fig. 6e). This trend was not observed in B6 *Nnmt* KO mice harboring a 35-bp deletion, which was sufficient to abolish MNAM (Supplementary Fig. 6b–e).

The suppression of the uracil accumulation was correlated with the expression pattern of *Cad* (Fig. 5f). *Nnmt* KO suppressed *Cad* upregulation but could rescue neither *Ass1* downregulation nor *Upp1* upregulation. These results suggested that uracil was accumulated due to enhanced expression of *Cad* and urea cycle dysregulation rather than *Upp1* upregulation. It is thus likely that NNMT modulates uracil biogenesis independently of UPP1-dependent uracil production from uridine.

Next, we explored how deletion of *Nnmt* rescued the urea cycle dysfunction. Among NAM, SAM, MNAM, and SAH, only MNAM was increased in the livers upon 4T1 breast cancer transplantation (Fig. 1d). Moreover, MNAM was almost completely depleted via *Nnmt* deletion (Fig. 2c, d). We thus hypothesized that MNAM at least in part mediates the NNMT-dependent changes. To test this hypothesis, we injected MNAM into cancer-free mice and measured a series of NNMT-affected metabolites. Injection of 250 mg/kg MNAM for 12 days significantly increased MNAM, me4PY, and me2PY in the liver (Fig. 5g and Supplementary Fig. 6f) as well as arginine and arginosuccinate (Fig. 5h). In contrast, MNAM did not affect uracil (Fig. 5h). Altogether, these results indicated that MNAM mediates at least part of the observed cancer-induced abnormalities (i.e., urea cycle dysregulation) in liver metabolism. Given the partial effects of MNAM, we expect that other NNMT-related metabolites such as SAM and NAD also contribute to the liver abnormalities depending on 4T1 and NNMT that we detected via multi-omics analyses. The detailed molecular mechanisms by which NNMT affects liver metabolism in the cancer-bearing condition remain to be elucidated.

Considering both the transcriptional and metabolic data, *Nnmt* appeared to be critical for the metabolic regulation of liver function in the cancer state. If this were true then *Nnmt* disruption should not only restore gene expression and metabolism within the liver of cancer-bearing mice but should also ameliorate overt phenotypes such as weight loss. Indeed, *Nnmt* KO significantly rescued weight loss seen in WT mice: 4T1 breast cancer transplantation massively reduced body weight 14 days after cancer transplantation, which was significantly rescued by *Nnmt* KO (Fig. 5i) while food intake was unaltered (Supplementary Fig. 7a). Furthermore, *Nnmt* KO partially rescued cancer-dependent decrease of the voluntary wheel-running activity (Fig. 5j). We also noted that *Nnmt* KO partially rescued the cancer-induced atrophy of adipose tissues (Supplementary Fig. 7b). 4T1 breast cancers massively reduced adipose tissues on day 14 after transplantation, which was significantly ameliorated in *Nnmt* KO mice. At this time point, the muscle mass was not significantly affected (Supplementary Fig. 7c). Furthermore, it appeared that LLC-bearing mice exhibited weaker systemic phenotypes compared to 4T1-bearing mice on day 14 after transplantation (Supplementary Fig. 7d, e), which was correlated with the lesser *Nnmt* upregulation in the livers of LLC-bearing mice (Fig. 1b, c).

Together, these results suggest an important role of the liver NNMT pathway in cancer cachexia at the multi-organ level.

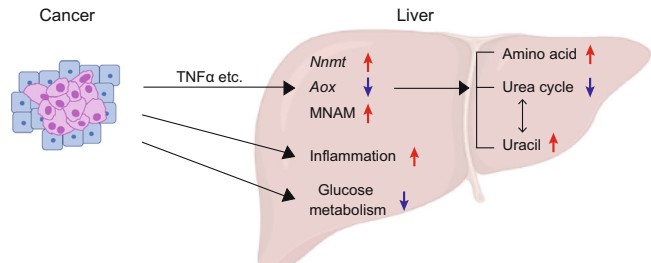

**Fig. 6 Graphical summary.** Solid cancers increase NNMT and MNAM in the liver, consequently causing dysregulation of the urea cycle and uracil biogenesis. On the other hand, NNMT is not required for abnormalities in glucose metabolism and enhancement of inflammation.

There would be multiple possibilities that explain how the metabolic dysfunction is related to systemic phenotypes. We assume that the reduced physical activity might be owing to deregulated energy imbalance. For example, the massive loss of adipose tissues may cause the deletion of energy for physical activity. The causal relationship among these phenotypes needs to be further studied.

## Discussion

In the current study, we combined mouse genetics and multi-omics analyses to demonstrate that host NNMT is involved in cancer-induced metabolic dysfunctions within the liver (Fig. 6). We also found that MNAM at least in part mediates the NNMT-dependent abnormalities.

*Nnmt* is one of the host genes responsive to cancer transplantation. *Nnmt* is increased in the liver in various cancer models (Fig. 1b, c and Supplementary Fig. 1a), suggesting a general role of *Nnmt* in cancer-induced liver abnormalities. TNFα is at least in part responsible for the *Nnmt* upregulation in hepatocytes (Fig. 1g). At this point, it is unclear whether cancer-derived TNFα directly affects the liver in vivo since it is possible that immune cells activated in the presence of cancers could also be a source of TNFα[1]. Further experiments are warranted to elucidate the major source of TNFα in vivo.

Expression of *Nnmt* was well correlated with its product MNAM in in vivo cancer transplantation experiments, in vitro culture experiments, and *Nnmt* knockout experiments (Figs. 1b, d–h, 2c, d and Supplementary Fig. 6b–d). The strong dependency of MNAM on NNMT has been suggested in a series of the previous publications[10–14,16,17,27]. The accumulation of MNAM in the cancer-bearing condition is likely also owing to the down-regulation of AOX-dependent MNAM metabolism (Fig. 1d and Supplementary Fig. 3a). The steady-state amounts of SAM, NAM, SAH, and NAD were not largely dependent on *Nnmt* in the healthy livers (Fig. 2c and Supplementary Fig. 2b). SAM became dependent on *Nnmt* in the cancer-bearing condition where the major SAM consumer *Gnmt* was decreased (Fig. 2d and Supplementary Fig. 2c). Collectively, we concluded that NNMT is constitutively critical for MNAM biogenesis and important for SAM homeostasis in a context-dependent manner.

We demonstrated that 4T1 breast cancers caused accumulation of amino acids, urea cycle dysregulation, and enhanced uracil biogenesis in the liver, all of which genetically require *Nnmt* (Figs. 4, 5). We speculate a causal relationship between the accumulation of amino acids and urea cycle dysregulation. It is known that protein degradation is enhanced particularly in the muscle of cachexia animals, which represents a hallmark of cancer cachexia[1]. Amino acids produced by such massive protein degradation are probably excessive to the body. Related to this, a recent study shows that human cachexia patients accumulate

amino acids including lysine, leucine, valine, and so on, in serum[28]. Accumulation of amino acids in the liver is also observed in the Colon26 model as shown in the recent study[29]. Given that *Nnmt* KO mice did not accumulate a set of amino acids in the liver upon cancer transplantation (Fig. 4), NNMT-dependent urea cycle dysregulation is a possible cause of the amino acid accumulation in the cancer-bearing condition. The causal interaction among enhanced protein degradation, amino acid accumulation, and urea cycle dysfunction in cancer cachexia remains to be solved.

Urea cycle dysregulation and enhanced uracil biogenesis were also well correlated in our datasets (Fig. 5). The decrease of plasma urea suggested urea cycle dysregulation as downregulation (Fig. 5d). Urea cycle dysregulation was accompanied by *Ass1* downregulation and *Cad* upregulation (Fig. 5c, f). These changes together with an increase in aspartate are indicative of enhanced flux of aspartate towards uracil biogenesis. In fact, uracil was accumulated in the livers of 4T1 breast cancer-bearing mice (Fig. 5e). This condition could be reversed through deletion of *Nnmt* that normalized urea cycle dysregulation, *Cad* upregulation, and uracil accumulation (Fig. 5b–f). These results indicate that cancers reprogram host metabolism via NNMT toward uracil biogenesis rather than nitrogen disposal. It is also possible that such metabolic alteration is a compensatory response of the liver to dispose nitrogen in a different way. The mechanisms underlying how NNMT is involved in these abnormalities are currently unclear. Despite this, our data showed that MNAM is capable of affecting a series of urea cycle metabolites (Fig. 5g, h). This observation is in line with our data that 4T1 breast cancers actively accumulate MNAM through *Nnmt* upregulation and suppression of MNAM metabolism to me4PY and me2PY (Supplementary Figs. 1b, 3a). These results highlight the previously unappreciated roles of NNMT and MNAM in regulating the urea cycle and uracil biogenesis in the cancer-bearing condition.

Interestingly, urea cycle dysregulation and elevated uracil biogenesis in the livers of cancer-bearing mice are similar to those observed in cancer cells[33,34]. The previous studies indicated that cancer cells suppress the urea cycle and nitrogen utilization while promoting uracil biogenesis to support their proliferation[33,34]. At the core of this metabolic reprogramming is the downregulation of ASS1 that increases the aspartate availability[34]. This consequently activates CAD to enhance uracil biogenesis. Such metabolic dysfunction recently turned out to be prevalent in various types of cancers and even detectable in the bio-fluids of cancer patients[33]. We speculate that urea cycle dysregulation detected in the blood of cancer patients can be attributed to the liver abnormalities rather than the cancer mass as the authors pointed out in their paper[33]. Our study thus sheds light on the importance of rewired host metabolism in cancers.

Our multi-omics datasets also revealed the complex picture of altered methyl-donor balance in the livers of 4T1 breast cancer-bearing mice (Fig. 2d and Supplementary Fig. 5c). Maintenance of appropriate methyl-donor balance is critical for organisms, and dysfunction of this pathway is strongly associated with multiple diseases[36]. It has been demonstrated that GNMT is the most important factor for methyl-donor balance in the liver[26,27]. GNMT transfers a methyl group from SAM to glycine to produce sarcosine. In the cancer-bearing condition, the hepatic *Gnmt* is decreased, suggesting the decreased usage of SAM. Yet, 4T1 breast cancer transplantation did not increase SAM in the WT livers. This could be attributed to the increase of *Nnmt* in the liver, which promotes SAM consumption (Supplementary Fig. 5c). Indeed, in the absence of *Nnmt*, the livers of 4T1 breast cancer-bearing mice accumulated SAM (Fig. 2d).

We also found an additional factor nicotinic acid (NA) that might be involved in methyl-donor balance in the liver. NA

accepts a methyl group to generate methylnicotinic acid also known as trigonelline (Supplementary Fig. 5c)[30]. Trigonelline is one of the most elevated metabolites in the livers of sham-operated *Nnmt* knockout mice (Supplementary Fig. 5a, b), suggesting that the yet-to-be-identified trigonelline biogenesis pathway consumed SAM in the absence of *Nnmt*. In the WT livers of 4T1 breast cancer-bearing mice, not only MNAM but also trigonelline was elevated (Supplementary Fig. 5a, b). These results point out the possibility that the increases of MNAM and trigonelline compensate for the *Gnmt* downregulation. In the presence of 4T1 breast cancer in *Nnmt* KO mice, the trigonelline pathway seemed insufficient to receive methyl-groups from excess SAM, finally resulting in the accumulation of SAM along with methyl-donor imbalance (Fig. 2d, e and Supplementary Fig. 2c). It is likely that such alterations in SAM homeostasis might play roles in liver abnormalities in cancer-bearing conditions. This finding highlighted the strength of an unbiased multi-omics approach to understand the entire picture of cancer-induced metabolic dysfunction in the host.

This study also uncovered abnormalities that do not require NNMT. *Nnmt* KO did not rescue liver inflammation in any significant way (Figs. 3, 6). Neither UPP1-dependent uracil production nor massive downregulation of glucose metabolism was dependent on NNMT (Fig. 5f and Supplementary Fig. 4). These results deliver two messages. First, we were able to classify host abnormalities based on a host factor (NNMT in this case). This shows the strengths of our strategy in dissecting the complex host pathophysiology in cancers. Second, the identification of host factors involved in NNMT-independent phenomena will be critical to establish therapeutics that efficiently ameliorate cancer's adverse effects on the host.

In summary, we identified NNMT as a host factor that mediates cancer-induced rewiring of nitrogen homeostasis in the liver (Fig. 6). This study paves the way toward deeper understanding of the mechanisms underlying host pathophysiology in cancers.

## Methods

**Mice**. All animal experiment protocols were approved by the Animal Care and Use Committee of Advanced Telecommunications Research Institute International (April 2014 to October 2018) and that of Kyoto University (October 2018 to current).

Mice were housed in a 12-h light/dark paradigm with food (CE-2, CLEA Japan, Inc., Tokyo, Japan (https://www.clea-japan.com/products/general_diet/item_d0030)) and water available *ad libitum*. Mice were randomly assigned to different experimental groups. No blinding was done. WT mice were purchased from Japan SLC Inc. (Hamamatsu, Japan).

**Generation of KO mice**. BALB/c *Nnmt* KO mice were generated as described previously[37]. Briefly, two independent guide RNAs (gRNAs) were cloned into the pX330 (#42230, Addgene, MA, USA). The sequence-validated vectors (FASMAC, Kanagawa, Japan) were injected into fertilized eggs of BALB/c mice (Transgenic, Fukuoka, Japan). The obtained founder (F0) mice were crossed with WT to obtain F1 mice. For generating C57BL/6 N KO mice, in vitro fertilized mouse eggs from 4-week-old females were harvested and stocked until use[38]. Fertilized eggs were thawed and electroporated using CUY-EDIT II (BEX, Tokyo, Japan) (amplitude 25 V, duration 3 msec., interval 97 msec. for twice) with two independent gRNAs (25 ng/ml: FASMAC), crRNAs (25 ng/ml: FASMAC), tracrRNA (25 ng/ml: FAS-MAC), and purified recombinant Cas9 proteins (250 ng/ml: Thermo Fisher Scientific, MA, USA). The sequences of gRNAs used for BALB/c and C57BL/6 N were the same (Supplementary Data 1). On the next day after electroporation, eggs at the two-cell stage were transplanted into the oviduct of pseudopregnant mice. F0 mice were crossed with WT and F1 mice were obtained for generating KO mice (≥F2).

**DNA extraction and genomic PCR**. Genomic DNAs were extracted from mouse tails. Tails were incubated with 90 μl of 50 mM NaOH (nacalai tesque, Kyoto, Japan) for 10 min at 95 °C. Ten microliters of 1 M Tris-HCl pH 8.0 (nacalai tesque) was then added to the reaction, followed by centrifugation at 12,000 rpm ($15{,}941 \times g$) for more than 10 min. The resulting supernatant were subjected to genomic PCR. Genomic PCR for genotyping were performed using KOD FX-Neo (TOYOBO, Osaka, Japan). The primers used are listed in Supplementary Data 1.

Representative pictures from more than 100 genotyped animals are shown in Fig. 2b and Supplementary Fig. 6c.

**Cell lines.** 4T1 mouse breast cancer cell line[6] and Colon26 mouse colon cancer cell line (a gift from Dr. T. Nojiri) were cultured in RPMI1640 (nacalai tesque) supplemented with 10% fetal bovine serum (nacalai tesque), 1% penicillin/streptomycin (nacalai tesque) in a 5% $CO_2$ tissue culture incubator at 37 °C. ID8 $Trp53^{-/-}$ F3 mouse ovarian cancer cell line (simply referred to as ID8-F3 in the manuscript) was provided by Dr. McNeish and cultured as described previously[39]. Mouse Lewis Lung Carcinoma (LLC) cell line (RCB0558) was obtained from the RIKEN BioResourse Center, Japan. LLC cells were cultured in DMEM low glucose (nacalai tesque) supplemented with 10% fetal bovine serum and 1% penicillin/streptomycin. AML12 cells were obtained from American Type Culture Collection (ATCC, VA, USA: CRL-2254™) and cultured in DMEM/Ham's F-12 (nacalai tesque) containing 10% fetal bovine serum, 1% penicillin/streptomycin, 40 ng/ml DEX (nacalai tesque), and 1 × Insulin-Transferrin-Selenium (Gibco, MA, USA). All cells were cultured and maintained in a 5% $CO_2$ tissue culture incubator at 37 °C.

**Cancer transplantation.** About $2.5 \times 10^6$ 4T1, $1.5 \times 10^6$ Colon26, and $5 \times 10^6$ LLC cells were inoculated subcutaneously into the right flank of 8–10-week-old BALB/c females, BALB/c males, and C57BL/6 N males, respectively. About $5 \times 10^6$ ID8-F3 cells were injected into the abdominal cavity of 8–10-week-old C57BL/6 N females. Mice were sacrificed at 7 and 14 days post-transplantation (4T1), 14 days post-transplantation (Colon26 and LLC), and 42 days post-transplantation (ID8-F3).

**RNA isolation and qPCR.** Mouse livers were crushed in liquid nitrogen and homogenized with Trizol reagent (Thermo Fisher Scientific). Total RNA was extracted from the homogenized supernatant using RNeasy Mini Kit (Qiagen, Venlo, Netherlands) according to the manufacturer's instruction. About 54–600 ng of total RNAs were reverse-transcribed with Transcriptor First Strand cDNA synthesis kit (Roche, Switzerland). qPCR experiments were performed using the StepOnePlus qPCR system (Applied Biosystems, CA, USA) and SYBR Green Master Mix (Roche, Basal, Switzerland). *Gapdh* and *B2m* were used as internal control. The primers used in these experiments are listed in Supplementary Data 1.

**Metabolite analyses.** In Figs. 1d, 2c–e and Supplementary Figs. 2b, d, 6d, the frozen livers were homogenized and metabolites were extracted in 50% methanol. Samples were deproteinized using 50% acetonitrile and completely evaporated. The resulting pellets were dissolved in Milli-Q water, followed by filtration with a 0.22 μm polyvinylidene fluoride (PVDF) filter (Millipore, MA, USA). Protein was extracted for normalization from the precipitates after metabolite extraction. The precipitates were washed in acetone and protein was extracted with 0.1 N NaOH at 95 °C for 5 min. The protein concentration was measured using a BCA Protein Assay kit (Thermo Fisher Scientific). Measurement of metabolites was performed using ultra-high-performance liquid chromatography equipped with tandem mass spectrometry, TQD (Shimadzu, Kyoto, Japan) as previously described[40]. For measurements, extracted samples were diluted in an equal volume of Milli-Q water. Each sample was then injected, and the concentration was calculated based on the standard curve obtained from the serial dilution of the standard solution for each metabolite.

In Figs. 1f, h, 4, 5 and Supplementary Figs. 1b, 2a, 4, 5, 6f, metabolites were extracted from the frozen livers (~5 mg), plasma (50 μL), or AML12 cells (~6 × 10⁵ cells/well (six-well plate)) using the Bligh and Dyer's method[41] with some modifications. Briefly, each sample was mixed with 1 mL of cold methanol containing 10-camphorsulfonic acid (1.5 nmol) and piperazine-1,4-bis (2-ethanesulfonic acid) (PIPES, 1.5 nmol) as internal standards for mass spectrometry-based metabolomic analysis. The samples were vigorously mixed by vortexing for 1 min followed by 5 min of sonication. The extracts were then centrifuged at 16,000 × *g* for 5 min at 4 °C, and the resultant supernatant (400 μL) was collected. After mixing 400 μL of supernatant with 400 μL of chloroform and 320 μL of water, the aqueous and organic layers were separated by vortexing and subsequent centrifugation at 16,000 × *g* and 4 °C for 5 min. The aqueous (upper) layer (500 μL) was transferred into a clean tube. After the aqueous layer extracts were evaporated under vacuum, the dried extracts were stored at −80 °C until the analysis of hydrophilic metabolites. Prior to analysis, the dried aqueous layer was reconstituted in 50 μL of water. Two liquid chromatography high-resolution tandem mass spectrometry (LC/MS/MS) methods for hydrophilic metabolite analysis were employed as described previously[42,43]. Anionic polar metabolites (i.e., organic acids, sugar phosphates, nucleotides, etc.) were analyzed via ion chromatography (Dionex ICS-5000⁺ HPIC system, Thermo Fisher Scientific) with a Dionex IonPac AS11-HC-4 μm column (2 μm i.d. × 250 mm, 4 μm particle size, Thermo Fisher Scientific) coupled with a Q Exactive, high-performance benchtop quadrupole Orbitrap mass spectrometer (Thermo Fisher Scientific) (IC/MS/MS). Cationic polar metabolites (i.e., amino acids, bases, nucleosides, NAM, SAM, MNAM, SAH, me2PY, me4PY, etc.) were analyzed via liquid chromatography (Nexera X2 UHPLC system, Shimadzu) with a Discovery HS F5 column (2.1 mm i.d. × 150 mm, 3 μm particle size, Merck) coupled with a Q Exactive instrument (PFPP-LC/MS/MS). The two analytical platforms for hydrophilic metabolite analysis were controlled using LabSolutions, version 5.80 (Shimadzu) and Xcalibur

4.2.47 (Thermo Fisher Scientific). The quantitative content of the hydrophilic metabolites was calculated using peak area relative to the internal standards of the HRMS precursor (PIPES for $[M + H]^+$ metabolites and 10-camphorsulfonic for $[M–H]^-$ metabolites) and then corrected for the total amount of each sample. In Fig. 1f, a triple quadrupole mass spectrometer (LC-MS-8060NX; Shimadzu) was also used.

**Culturing AML cells with the 4T1-conditioned media.** About $1 \times 10^6$ 4T1 cells were cultured in 10 cm dishes for 48 h and the culture supernatant was collected. The culture supernatant was stored in the 4T1-conditioned media at 4 °C until use. About $1 \times 10^5$ AML cells per well were cultured in a six-well plate for 24 h, and then the media was switched to the 4T1-conditioned media. After 24 h, the treated AML12 cells were collected and subjected to qPCR and LC-MS/MS.

**Treating AML12 cells with TNFα.** About $0.25 \times 10^5$ AML cells per well were cultured in a 24-well plate for 24 h. The media was then switched to the bovine-serum free media, and TNFα was added at the concentration of 20 ng/ml or 200 ng/ml (Roche). After 24 h, the treated AML12 cells were collected and subjected to qPCR and LC-MS/MS.

**Blood collection and plasma urea measurement.** Whole blood samples were collected from the hearts using heparin-coated syringes (heparin-Na, 1000 U/ml; nacalai tesque) and centrifuged at 1500 × *g* for 20 min at 4 °C. The obtained supernatant (plasma) samples were stored at −80 °C and plasma urea was measured at ORIENTAL YEAST CO., LTD, Tokyo, Japan.

**TNFα measurement.** 4T1 cells were cultured in a 10 cm dish to be confluent and then the supernatant was collected. After centrifugation at 1000 rpm (161 × *g*) for 5 min at room temperature, the resulting supernatant was collected and stored at −80 °C until measurements. TNFα was measured using Bio-Plex 200 and Mouse Cytokine TNF-α Set, Mouse Cytokine Standards Group I, and Reagent Kit V with Flat Bottom Plate (Bio-Rad, CA, USA).

**Monitoring of voluntary wheel running and food intake.** Mice were housed individually and acclimated to the mouse activity monitoring system (cFDM-300AS, MELQUEST, Toyama, Japan) for 1 week before the initiation of the experiment. About $2.5 \times 10^6$ 4T1 cells were inoculated subcutaneously into the right flank of 8–9-week-old BALB/c WT and *Nnmt* KO female mice. The wheel-running activity and food intake were measured automatically for 14 days using the cFDM-300AS system. Data collected at half-hour intervals were analyzed using Feedam-BMPC software (MELQUEST).

**Transcriptome analysis.** Total RNAs were extracted from the livers as described above. RNA-seq libraries were generated using the Illumina TruSeq Stranded mRNA preparation kit according to the manufacturer's instructions (Illumina, CA, USA). Sequencing experiments were performed with NovaSeq 6000 system (Illumina; Single End 100 bp). The obtained reads were mapped to the mouse genome mm10 using Illumina Eland with the default parameter setting. The obtained gene list with reads per million per kilobase (RPKM) scores were shown in Supplementary Data 2. RPKM scores from four replicates were averaged, and the ratio between 4T1-bearing and sham-operated conditions was calculated in WT and *Nnmt* KO. The ratio between sham-treated WT and *Nnmt* KO was also calculated. The obtained ratios were used to sort genes to find candidate differentially expressed genes (DEGs). DEGs were further subjected to gene ontology analyses using g:Profier[44].

**Statistical analysis and data visualization.** The sample size was empirically determined depending on the size effects. The number of animals was minimized as much as possible in light of animal ethics. In most cases, n = 4–5 was set as a threshold. Dot-plot representation of data was exploited to obtain insights into how the samples were distributed and thus into the extent of difference between the two groups.

Significant differences between the two groups were estimated using a two-tailed, paired, or unpaired Student's *t*-test. The false discovery rate was estimated by *q* value using Storey's method[45]. Graphs were drawn using Excel, GraphPad Prism 9, and R. Figure 6 was depicted using BioRender.

**Reporting Summary.** Further information on research design is available in the Nature Research Reporting Summary linked to this article.

## Data availability

Source data are provided as a source data file. The RNA-seq data have been deposited in the DNA Data Bank of Japan (DDBJ) under the accession code DRA011922. The metabolomics data have been deposited in Metabolomics Workbench (https://www.metabolomicsworkbench.org/) under the study IDs of ST002163 and ST002167. All other data are included as supplementary information.

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

## Acknowledgements

This work was supported by JSPS KAKENHI (17H06299, 18K15409, 18H04810, 20H03451, and 20H04842; S.K.), JST FOREST (JPMJFR2062; S.K.), JST Moonshot (JPMJMS2011-61; S.K.), JST ERATO Sato Live Bio-forecasting project (JPMJER1303), Caravel, Co., Ltd (S.K.), Ono Medical Research Foundation (S.K.), Takeda Science Foundation (S.K.), Mochida Memorial Foundation (S.K.), The Uehara Memorial Foundation (S.K.), Chubei Ito Foundation (S.K.), and Japan Foundation for Applied Enzymology (S.K.). This work was also supported by JSPS KAKENHI (19K05167; Y.I.), JST MIRAI (Y.I.), JSPS KAKENHI (JP21H04774; M. Miura), and AMED-Aging (JP21gm5010001; M. Miura). We thank Dr. Bryce Nelson, Dr. Pieter Bas Kwak, Dr. Hiroki Shibuya, Dr. Takeshi Watanabe, and Dr. Kosuke Yusa for critically reading the manuscript. We also thank Dr. Gen Kondoh, Dr. Hitomi Watanabe, Hitoshi Miyachi, and Satsuki Kitano for their help in mouse experiments.

## Author contributions

S.K conceived and supervised the project and wrote the paper. R.M. performed experiments, analyzed data, constructed figures, and wrote the paper. H.H. performed experiments and analyzed data. S.E., R.K., A.H., and M.Y. performed experiments. M.T, S. K., M.N., M. M., T.B., and Y.I. performed metabolites measurements. Y.S. performed RNA-seq experiments. H.K., J.H., and M. M. made a substantial contribution to the conception of this work. All authors provided intellectual input and reviewed the paper.

## Competing interests

The authors declare no competing interests.
