## [Peer Review File · Nature Communications]

Remote solid cancers rewire hepatic nitrogen metabolism via host nicotinamide-N-methyltransferaseEditorial Note: Parts of this Peer Review File have been redacted as indicated to maintain the confidentiality of unpublished data.

REVIEWER COMMENTS

Reviewer #1 (Remarks to the Author):

The study by Mizuno et al elucidates how systemic factors from tumors can reprogram hepatic metabolism through the deregulation of nicotinamide-N38 methyltransferase (NNMT). The authors show that tumor-bearing mice increase the expression of NNMT and its product MNAM in the liver, which is linked to altered urea cycle and uracil biogenesis. Deletion of NNMT abolishes MNAM and reduced cancer-induced weight loss and wheel-running activity. The authors conclude that cancers activate the NNMT pathway towards uracil biogenesis rather than nitrogen disposal through the urea cycle.

The study is interesting and focuses on an important area of systemic metabolism connecting the tumor, liver, and muscle, which is poorly understood. The knockout data is strong and establishes the role of NNMT in cancer-induced weight loss. However, the study pieces together multiple profiling results but fails to connect them to the phenotype. Additionally, there are several gaps in the study and a lack of mechanistic insights. Details are provided below.

- The study builds on many assumptions of systemic deregulation in the liver and weight loss, which were not substantiated. The authors failed to show how reduced urea synthesis and increased uracil synthesis in the liver are connected to the phenotype (bodyweight loss and activity), which is the central claim of the study.
- NNMT upregulation in the liver has been reported before in the same C26 model (PMID 9703363) so that observation is not novel.
- NNMT is highly upregulated in the C26 and 4T1 models but only modestly in LLC. All three models induce severe weight loss/cachexia. Is NNMT linked to the type of tumor and not cachexia?
- Urea excretion is higher in mice undergoing weight loss according to the literature, which does not match the profiling results presented here. The reasons are unclear.
- Is the weight loss from adipose tissue loss or muscle, and what is rescued in the knockout (adipose tissue/muscle loss)?
- How is the wheel-running activity connected to the metabolic phenotype?

Reviewer #2 (Remarks to the Author):

This is a highly original paper that will be worthy of publication after some thoughtful revision. The authors have shown that in response to particular types of transplanted malignancies, a mouse's liver metabolism is altered in a manner that induces expression of NNMT, driving production of meNAM, depression of the urea cycle and elevated production of uracil. Deletion of host NNMT abrogates some of the characteristic and damaging effects of tumor burden include depression of voluntary running. The authors completed a body of work here and described their work effectively without overinterpretation. However, in a revision they should make the following improvements:

1) After meNAM is formed, there is a lot of me4PY and me2PY formed. Ideally, the authors quantify these because their sum is closer to the sum of NNMT activity than just meNAM. Our group has described this, particularly in the context of NR supplementation, which also produces increased flux through NNMT.

2) Figures 5a and 6 leave quite a bit to be desired. If we are viewing the urea cycle as a nitrogen disposal mechanism, we can also view uracil formation as a nitrogen disposal system but really the nitrogen is coming from glutamine and the glutaminase activity of CAD. The authors could be much more helpful with better tracking of nitrogen and amino acids in their cartoons. Can the authors try to explain the metabolic changes as the host liver's homeostatic attempt to dispose of N in a different way? Do they think that there could be negative consequences of NNMT deletion or inhibition? Can they help the reader make sense of the physical activity phenotype?

Charles Brenner

Point-by-point responses to the reviewer's comments

Please find our point-by-point responses to the reviewer's comments as below.

Responses to reviewer #1

The study by Mizuno et al elucidates how systemic factors from tumors can reprogram hepatic metabolism through the deregulation of nicotinamide-N38 methyltransferase (NNMT). The authors show that tumor-bearing mice increase the expression of NNMT and its product MNAM in the liver, which is linked to altered urea cycle and uracil biogenesis. Deletion of NNMT abolishes MNAM and reduced cancer-induced weight loss and wheel-running activity. The authors conclude that cancers activate the NNMT pathway towards uracil biogenesis rather than nitrogen disposal through the urea cycle.

The study is interesting and focuses on an important area of systemic metabolism connecting the tumor, liver, and muscle, which is poorly understood. The knockout data is strong and establishes the role of NNMT in cancer-induced weight loss. However, the study pieces together multiple profiling results but fails to connect them to the phenotype. Additionally, there are several gaps in the study and a lack of mechanistic insights. Details are provided below.

We thank the reviewer #1 for providing the insightful comments on our manuscript. We hope that our revisions satisfy the reviewer's concerns.

1) The study builds on many assumptions of systemic deregulation in the liver and weight loss, which were not substantiated. The authors failed to show how reduced urea synthesis and increased uracil synthesis in the liver are connected to the phenotype (bodyweight loss and activity), which is the central claim of the study.

It has been known that cancers cause metabolic dysfunction in the liver, adipose and muscle degradation, leading to weight loss. Deletion of *Nnmt*, one of the dysregulated genes in the liver, rescues cancer-induced urea cycle dysfunction and substantially ameliorates weight loss (Fig. 5). In addition, this revision demonstrates that NNMT plays a role in the adipose loss (Extended Data Fig. 7b). Collectively, we establish a novel, NNMT-dependent link between metabolic abnormalities and systemic phenotypes. This is our central claim. However, as pointed out by this reviewer, the mechanism behind this link is unclear and would be definitely interesting to investigate further in the future studies.

Regarding this comment, we have added the following sentences.

“Together, these results suggest an important role of the liver NNMT pathway in cancer cachexia at the multi-organ level. There would be multiple possibilities that explain how the metabolic dysfunction is related to systemic phenotypes. We assume that the reduced physical activity might be owing to deregulated energy imbalance. For example, the massive loss of adipose tissues may cause depletion of energy for the physical activity. The causal relationship among these phenotypes needs to be further studied”. (Line 314-319 in the revised manuscript)

2) NNMT upregulation in the liver has been reported before in the same C26 model (PMID 9703363) so that observation is not novel.

We are aware of this paper and had already cited it in introduction of the original manuscript (Line 76-78 in the revised manuscript).

According to this comment we have also added the following sentence in the revised manuscript.

“The Colon26 data were consistent with the previous finding²¹.”
(Line 100 in the revised manuscript)

3) NNMT is highly upregulated in the C26 and 4T1 models but only modestly in LLC. All three models induce severe weight loss/cachexia. Is NNMT linked to the type of tumor and not cachexia?

We appreciate this comment. As noted by this reviewer, all three models induce cachexic phenotypes but require different duration to fully induce cachexia. In our hands, LLC takes longer time than 4T1 to induce cachexia. In this study, we investigated the adipose and muscle loss on day 14 after LLC transplantation (Extended Data Fig. 7d-e). At this time point, in contrast to the 4T1 data (Extended Data Fig. 7b-c), neither adipose tissues nor muscle tissues were decreased by LLC transplantation (Extended Data Fig. 7d-e), and *Nnmt* was only modestly up-regulated in this model (Fig. 1c). Thus, the degree of NNMT up-regulation correlates the degree of cachexic phenotypes. To clarify this point, the following sentences have been added in the revised manuscript.

“Furthermore, it appeared that LLC-bearing mice exhibited weaker systemic phenotypes compared to 4T1-bearing mice on day 14 after transplantation (Extended Data Fig. 7d-e), which was correlated with the lesser *Nmmt* up-regulation in the livers of LLC-bearing mice (Fig. 1b-c).” (Line 310-313 in the revised manuscript)

4) Urea excretion is higher in mice undergoing weight loss according to the literature, which does not match the profiling results presented here. The reasons are unclear.

We appreciate this interesting suggestion. To investigate whether the amount of urea in the urine is correlated with the urea cycle dysfunction in the liver, we measured urea in the urine, finding that the urea level did not increase but rather tended to decrease in the urine of 4T1-bearing mice compared to sham mice (Fig. R1). Multi-omics analyses revealed that the urea cycle is suppressed while uracil production is enhanced in the liver (Fig. 5). Thus, there is no contradiction regarding the urea level.

Fig. R1 Measurements of urea in the urine

Urea in the urine measured on day14 after 4T1 transplantation. $n = 4$.

Averaged fold change data to the sham group are presented as the mean \pm SEM.

5) Is the weight loss from adipose tissue loss or muscle, and what is rescued in the knockout (adipose tissue/muscle loss)?

This point turned out to be highly important. We found that *Nmmt* KO rescued the loss of adipose tissues in the 4T1 model (Extended Data Fig. 7b). Related to this, we have now added the following sentence in the revised manuscript.

“We also noted that *Nnmt* KO partially rescued the cancer-induced atrophy of adipose tissues (Extended Data Fig. 7b). 4T1 breast cancers massively reduced adipose tissues on day 14 after transplantation, which was significantly ameliorated in *Nnmt* KO mice. At this time point, the muscle mass was not significantly affected (Extended Data Fig. 7c).” (Line 306-309 in the revised manuscript)

6) How is the wheel-running activity connected to the metabolic phenotype?

There would be multiple possibilities to explain how the metabolic phenotypes are related to the wheel-running activity. The massive loss of adipose tissues observed in cancer-bearing WT mice may reflect energy imbalance in systemic metabolism, which would underlie the reduced wheel-running activity. However, addressing this issue is beyond scope of this manuscript. The causal relationships among the observed molecular and systemic phenotypes need to be clarified further. We have corrected our original description regarding this point as follows.

“Together, these results suggest an important role of the liver NNMT pathway in cancer cachexia at the multi-organ level. There would be multiple possibilities that explain how the metabolic dysfunction is related to systemic phenotypes. We assume that the reduced physical activity might be owing to deregulated energy imbalance. For example, the massive loss of adipose tissues may cause depletion of energy for the physical activity. The causal relationship among these phenotypes needs to be further studied”. (Line 314-319 in the revised manuscript)

Responses to reviewer #2

This is a highly original paper that will be worthy of publication after some thoughtful revision. The authors have shown that in response to particular types of transplanted malignancies, a mouse's liver metabolism is altered in a manner that induces expression of NNMT, driving production of meNAM, depression of the urea cycle and elevated production of uracil. Deletion of host NNMT abrogates some of the characteristic and damaging effects of tumor burden include depression of voluntary running. The authors completed a body of work here and described their work effectively without overinterpretation. However, in a revision they should make the following improvements:

We appreciate the positive and encouraging comments from this reviewer.

1) After meNAM is formed, there is a lot of me4PY and me2PY formed. Ideally, the authors quantify these because their sum is closer to the sum of NNMT activity than just meNAM. Our group has described this, particularly in the context of NR supplementation, which also produces increased flux through NNMT.

We appreciate this important suggestion. We have now measured me4PY and me2PY levels and obtained a couple of interesting data.

First, we found that me4PY and me2PY were completely abolished in the liver (Extended Data Fig. 2a), plasma (Fig. R2a), and urine (Fig. R2b) of *Nnmt* KO mice, validating that me4PY and me2PY productions are absolutely dependent on NNMT.

Fig. R2 Measurements of me2PY and me4PY in the plasma and urine

a. LC-MS/MS analysis for me2PY and me4PY in the plasma of WT and *Nnmt* KO mice. $n = 3$.

b. LC-MS/MS analysis for me2PY and me4PY in the urine of WT and *Nnmt* KO mice. $n = 3$.

Averaged fold change data to WT are presented as the mean \pm SEM.

Curiously, however, me4PY and me2PY levels were not elevated in the livers of 4T1-bearing mice, despite the increase of MNAM level (Fig. 1d and Extended Data Fig. 1b). To ask how this happens, we mined our RNA-seq data and found that the expression of *Aox1* and *Aox3*, the genes encoding enzymes that produces me4PY and me2PY from MNAM, was reduced (Extended Data Fig. 3a). Therefore, 4T1 breast cancers increase MNAM through up-regulation of NNMT and down-regulation of the production of me4PY and me2PY.

In contrast, upon direct injection of MNAM into healthy mice, MNAM, me4PY, and me2PY levels were all elevated in the liver (Fig. 5g and Extended Data Fig. 6f). This result together with the liver data suggest that 4T1 breast cancers rewire liver metabolism towards the higher accumulation of MNAM (Fig. 6).

We have added the following sentences in the revised manuscript and added the following references.

“In addition, it is known that MNAM is further metabolized into me4PY (*N*-methyl-4-pyridone-3-carboxamide) and me2PY (*N*-methyl-2-pyridone-5-carboxamide) by aldehyde oxidase (AOX)^{10,22}. Yet, 4T1 transplanted did not affect these two metabolites (Extended Data Fig. 1b).” (Line 111-114 in the revised manuscript)

“During this analysis, we found that *Aox1* and *Aox3* were down-regulated in the livers of 4T1-bearing mice (Extended Data Fig.3a). As described earlier, AOX proteins metabolize MNAM into me4PY and me2PY^{10,22,23}. These together suggested that 4T1 breast cancers increase MNAM not only via NNMT up-regulation but also via down-regulation of *Aox* genes (i.e., suppression of MNAM degradation).” (Line 173-177 in the revised manuscript)

“Injection of 250 mg/kg MNAM for 12 days significantly increased MNAM, me4PY, and me2PY in the liver (Fig. 5g and Extended Data Fig. 6f) as well as arginine and arginosuccinate (Fig. 5h).” (Line 288-290 in the revised manuscript)

“The accumulation of MNAM in the cancer-bearing condition is likely also owing to the down-regulation of AOX-dependent MNAM metabolism (Fig 1d and Extended Data Fig. 3a).” (Line 334-336 in the revised manuscript)

“This observation is in line with our data that 4T1 breast cancers actively accumulate MNAM through *Nnmt* up-regulation and suppression of MNAM metabolism to me4PY and me2PY (Extended Data Fig. 1b, 3a).” (Line 368-390 in the revised manuscript)

2) Figures 5a and 6 leave quite a bit to be desired. If we are viewing the urea cycle as a nitrogen disposal mechanism, we can also view uracil formation as a nitrogen disposal system but really the nitrogen is coming from glutamine and the glutaminase activity of CAD. The authors could be much more helpful with better tracking of nitrogen and amino acids in their cartoons. Can the authors try to explain the metabolic changes as the host liver's homeostatic attempt to dispose of N in a different way? Do they think that there could be negative consequences of NNMT deletion or inhibition? Can they help the reader make sense of the physical activity phenotype?

We appreciate the series of insightful comments on our manuscript.

We have amended Fig.5a and 6, and have added Extended Data Fig. 6a for more detailed explanation of the metabolic pathways we discuss. We tried better to allow readers to track nitrogen and amino acids. Furthermore, aspartate and glutamine are featured as nitrogen sources.

We also have added the following sentence to discuss the alternative possibility that the observed metabolic changes could be an attempt to dispose of nitrogen in a different way.

“It is also possible that such metabolic alteration is a compensatory response of the liver to dispose nitrogen in a different way.” (Line 364-366 in the revised manuscript)

[Redacted]

[Redacted]

Regarding the final point, there would be multiple possibilities to explain how the metabolic phenotypes are related to the wheel-running activity. We hypothesize that the reduced physical activity might be owing to deregulated energy imbalance caused by cancers. For example, the massive loss of adipose tissues may cause the lack of energy for mice to move around. However, addressing this issue is beyond scope of this manuscript. The causal relationships among the observed molecular and systemic phenotypes need to be clarified further. We have corrected our original description regarding this point as follows.

“Together, these results suggest an important role of the liver NNMT pathway in cancer cachexia at the multi-organ level. There would be multiple possibilities that explain how the metabolic dysfunction is related to systemic phenotypes. We assume that the reduced physical activity might be owing to deregulated energy imbalance. For example, the massive loss of adipose tissues may cause depletion of energy for the physical activity. The causal relationship among these phenotypes needs to be further studied”. (Line 314-319 in the revised manuscript)

Other modifications

We have added Mayuko Yoda and Ayano Harata as co-authors according to their contributions to our revisions. Moreover, we have revised the original Fig. 1f. In the original figure that harbored four replicates, we concluded that 4T1-supernatant elevated MNAM while SAM/NAM/SAH were unaltered. During the revision, we further added four new replicates, finding that 4T1-supernatant reduced SAM (Fig. 1f). This experiment was done using a LC-MS/MS newly introduced to our lab. The method section was accordingly revised.

“We found that 4T1-conditioned media was capable of up-regulating *Nnmt* expression in AML12 (Fig. 1e). This was accompanied by concomitant increase of MNAM and decrease of SAM (Fig. 1f).” (Line 120 in the revised manuscript)

REVIEWERS' COMMENTS

Reviewer #1 (Remarks to the Author):

The authors have addressed my concerns and the revised manuscript has improved.

Reviewer #2 (Remarks to the Author):

The manuscript has been effectively revised and is appropriate for publication in Nature Comms.

Point-by-point responses to the reviewer's comments

Reviewer #1 (Remarks to the Author):

The authors have addressed my concerns and the revised manuscript has improved.

We thank this reviewer for helping us to improve our manuscript.

Reviewer #2 (Remarks to the Author):

The manuscript has been effectively revised and is appropriate for publication in Nature Comms.

We appreciate this reviewer for helping us to revise our manuscript effectively.